# A Physics-Inspired Deep Learning Framework for an Efficient Fourier Ptychographic Microscopy Reconstruction under Low Overlap Conditions

**DOI:** 10.3390/s23156829

**Published:** 2023-07-31

**Authors:** Lyes Bouchama, Bernadette Dorizzi, Jacques Klossa, Yaneck Gottesman

**Affiliations:** 1Samovar, Télécom SudParis, Institut Polytechnique de Paris, 91120 Palaiseau, France; bernadette.dorizzi@telecom-sudparis.eu (B.D.); yaneck.gottesman@telecom-sudparis.eu (Y.G.); 2TRIBVN/T-Life, 92800 Puteaux, France; jklossa@tribvn.com

**Keywords:** Fourier Ptychographic Microscopy, deep learning, deep image prior

## Abstract

Two-dimensional observation of biological samples at hundreds of nanometers resolution or even below is of high interest for many sensitive medical applications. Recent advances have been obtained over the last ten years with computational imaging. Among them, Fourier Ptychographic Microscopy is of particular interest because of its important super-resolution factor. In complement to traditional intensity images, phase images are also produced. A large set of N raw images (with typically N = 225) is, however, required because of the reconstruction process that is involved. In this paper, we address the problem of FPM image reconstruction using a few raw images only (here, N = 37) as is highly desirable to increase microscope throughput. In contrast to previous approaches, we develop an algorithmic approach based on a physics-informed optimization deep neural network and statistical reconstruction learning. We demonstrate its efficiency with the help of simulations. The forward microscope image formation model is explicitly introduced in the deep neural network model to optimize its weights starting from an initialization that is based on statistical learning. The simulation results that are presented demonstrate the conceptual benefits of the approach. We show that high-quality images are effectively reconstructed without any appreciable resolution degradation. The learning step is also shown to be mandatory.

## 1. Introduction

Two-dimensional observation of biological samples at hundreds of nanometers resolution or even below is of high interest for many research activities such as fundamental biology, drug discovery, or other medical applications. This has motivated numerous studies in optical microscopy to push its actual limits regarding the delivered images’ ultimate resolution caused by light diffraction. Additional access to information such as sample optical thickness ($2D^{1/2}$) or even a complete 3D representation of the sample is also highly desired.

In such context, different computational microscopy approaches have made important progress over the last decade [1,2]. For these approaches, the image of the sample is obtained after solving an inverse problem. Most of available methods exploit multiple measurements of the sample under different experimental conditions [3,4,5,6]. Acquired data (or raw images) are then exploited computationally to reconstruct the information that is not directly accessible. Among available techniques, Fourier Ptychographic Microscopy (FPM) is of particular interest [7,8]. It is based on a conventional optical microscope with few modifications in the employed illumination source. An impressive super-resolution factor up to ∼5 is accessible (demonstrated experimentally) using 225 raw images together with a quantitative assessment of absorption and local optical thickness of samples. Hence, such microscopy is a strong candidate for different applications. In addition, the detection of rare events is facilitated by the intrinsic increase in the microscope’s space-bandwidth product (SBP).

Since the FPM acquisition time is proportional to the number of raw images that are captured, algorithmic approaches enabling good image reconstructions with few images are highly desirable. The first proposed method, from Waller’s group [9,10] consisted in multiplex coding illumination. In this approach, the specimen is simultaneously probed with different angular plane waves. The different regions in Fourier domain that are captured by the camera in a single shot are then separated with a sort of source separation algorithm embedded in the reconstruction algorithm. Such method has two advantages: the number of images that needs to be recorded for a complete coverage of full targeted spectral region is much reduced; the total optical power used to probe the sample is increased in proportion with the number of LEDs used simultaneously, leading to a possible reduction in the camera exposure time. However, the exact consequences of such coding on reconstructed image quality is not yet clear since the information acquired with a given camera dynamic needs to be distributed over large spectral regions. Alternative approaches [11,12,13,14] have also been investigated where the reconstruction process is obtained from training a deep neural network (DNN). Here, the training process aims at approximating a general underlying relation between the raw captured images and reconstructed ones. Reconstruction in a deep-learning framework has been mainly motivated by many additional foreseen advantages such as faster reconstructions brought by GPUs in feed-forward calculations (inference), or better noise tolerances. Successful DNN reconstructions have already been obtained when a large number of raw images are used. However the previous published results that have been obtained experimentally or with simulations are not yet completely satisfactory when only a few raw images are exploited. For example, artifacts often appear in the high frequencies of reconstructed images although perceptually close to images reconstructed with the original FPM method.

The situation is about to change with the advent of deep image prior (DIP) networks [15,16]. In such a scheme, a neural network is used as a prior to solve inverse problems. One of the remarkable specificities of DIP is that it does not exploit any prior training data. Image characteristics are captured by the structure of a convolutional image generator rather than by any previously learned capabilities. In practice, the neural architecture introduces some regularization into the classic inversion optimization scheme. DIP has been proved to be successful in many imaging application fields such as noise reduction, super-resolution, and inpainting. More recently, image reconstruction with improved axial resolution has been obtained in the field of optical diffraction tomography [17,18]. However, and in contrast to the training approach, DIP approaches lack generalization capabilities since the solution is specific to each image considered.

In this paper, we address the difficult problem of FPM reconstruction with few raw images in a deep-learning context. The method that we propose will go beyond the two previous trends; more precisely, we here introduce a general method based on DIP formalism. In the first step, the DIP network is trained for reconstruction. In the second step, its weights are further optimized in order to obtain a precise estimation of the sought solution that is specifically attached to a given set of raw images. The DIP network is fully physics-informed across the two steps in the sense that the exact forward model of FPM image formation and light-sample interaction is introduced in its loss function [19]. The principle of such a reconstruction scheme is demonstrated with the help of simulations. Reconstruction with a super-resolution factor ∼5 with no appreciable artifact in the high frequencies is achieved with only 37 raw images. A particular microscope illumination LED matrix pattern is considered in the simulations. Its spatial arrangement has been chosen in order to pave the Fourier space with as low a number of LEDs as possible. The approach is general and applicable to other illumination patterns such as the classic rectangular LED board. It can also open interesting perspectives for FPM reconstruction under multiplex-coded illumination to reduce further the number of captured images that are required.

This paper is organized as follows. The basic principle of FPM is briefly recalled in Section 2. Emphasize is put on the reconstruction problem when only a few low-resolution images are acquired. It is the key problem that is addressed and solved in Section 3 with the proposed cDIP-LO architecture. To this end, a unified DNN model is introduced, allowing both learning and optimization steps. In Section 4, the performance of the complete model is evaluated and discussed. With the help of simulation results, we demonstrate that both low and high frequencies are correctly reconstructed. The FPM configuration that is considered is associated with a super-resolution factor ∼5. The number of LR images exploited is reduced from 130 (overlap ∼60%) to 37 (overlap ∼10%). Section 5 is devoted to the conclusion and perspectives of the work.

## 2. Fourier Ptychographic Microscopy Principle

FPM relies on an optical microscope setup [7,8,20] in which the traditional illumination source is replaced by an LED matrix array (see Figure 1a). Each LED of the matrix is assimilated to a quasi-monochromatic punctual source [21]. Its distance to the sample is far enough to approximate the incident light with a plane wave whose wave-vector **k** varies with its spatial position. Changing the *i*th LED that is turned on permits us to probe the sample with varied $k^{i}$.

Let us denote $U_{in}^{i}\left(x,y\right)$ and $U_{out}^{i}\left(x,y\right)$, respectively, as the electric field that is incident on the sample and the electric field exiting from the sample for the *i*th. Under thin sample approximation, the light–matter interaction is modeled with a complex mask *T*. The relation between $U_{in}^{i}\left(x,y\right)$ and $U_{out}^{i}\left(x,y\right)$ becomes
(1)$$U_{out}^{i}\left(x,y\right)=U_{in}^{i}\left(x,y\right)\cdot T\left(x,y\right)$$
with $U_{in}^{i}\left(x,y\right)=Ae^{j(k_{x}^{i}\cdot \left(x\right)+k_{y}^{i}\cdot \left(y\right))}$. The image formation realized by the microscope at the camera plane consists in a low pass filtering of $U_{out}^{i}\left(x,y\right)$. The electric field $U_{cam}^{i}\left(x,y\right)$ at the camera plane becomes
(2)$$U_{cam}^{i}\left(x,y\right)=U_{out}^{i}\left(\frac{x}{g},\frac{y}{g}\right)*C\left(x,y\right)$$
where *g* represents the magnification factor of the objective lens, *C* the point spread function and ∗ the convolution product. In Fourier domain,
(3)$${\hat{U}}_{cam}^{i}\left(k_{x},k_{y}\right)=\hat{T}\left(g\left(k_{x}-k_{x}^{i}\right),g\left(k_{y}-k_{y}^{i}\right)\right)\cdot \hat{C}\left(k_{x},k_{y}\right)$$
where $\hat{T}$ and $\hat{C}$ represents the Fourier transform of *T* and *C*. For an objective lens that is aberration free, $\hat{C}\left(k_{x},k_{y}\right)=1$ in a region delimited by a disk of radius $r=\frac{2\pi }{\lambda }\cdot NA$ centered at (0, 0), and 0 outside, where NA represents the numerical aperture of the objective lens. Equation (Equation 3) defines the forward model that describes the image formation of the sample illuminated with the *i*th LED. It is interesting to observe that the *i*th LED enables the grabbing of the region in the spectral domain that is delimited by a disk of radius r and centered on $\left(k_{x}^{i},k_{y}^{i}\right)$. Figure 1b illustrates the position of these regions for the LED matrix that will be considered for simulations. It is composed of 37 LEDs arranged with a circular pattern.

Provided the phase is preserved, the many $U_{cam}^{i}$ fields can be readily assembled in the Fourier domain in order to recover the sample spectrum over a region that is much larger than the one directly accessible to the objective lens. This constitutes the heart of the synthetic aperture mechanism exploited by FPM. This permits us to recover the sample with an enhanced resolution. The super-resolution factor γ that is attained is theoretically only imposed by the LED matrix layout, and hence the spectral region that is paved.

The *i*th image captured by the camera is
(4)$$I^{i}\left(x,y\right)=|U_{cam}^{i}\left(x,y\right){|}^{2}$$

As a result, the phase is inevitably lost during the capture of the different LR images. It is hence mandatory to reconstruct the missing phase information. Different iterative approaches are available such as the Gershberg–Saxton’s [22,23], ePIE [24] or EPRY [25] algorithms. Let us denote $\hat{T}$ as an approximated solution and $I^{i}$ as the low-resolution image deduced after applying the direct model on $\hat{T}$ as indicated below:(5)$$\sqrt{I^{i}\left(x,y\right)}=|F^{-1}\left(\hat{T}\left(k_{x},k_{y}\right)\cdot \hat{C}\left(k_{x}-k_{x}^{i},k_{y}-k_{y}^{i}\right)\right)|$$
with $F^{-1}$ being the Fourier inverse transform. The searched solution is hence found by minimizing the error function.
(6)$$L=\sum_{i}{\left|\sqrt{I^{i}\left(x,y\right)}-\sqrt{I^{i}\left(x,y\right)}\right|}^{2}$$

With $I^{i}$ being the different measured images. The above-mentioned algorithms proceed iteratively and $\hat{T}$ is reconstructed progressively with descent-gradient calculations. These calculations operate by exploiting images $I^{i}$, sweeping LEDs *i* from 1 to N. The solution $\hat{T}$ is found after convergence. Super-resolved intensity and phase images are then calculated as the modulus and phase of the inverse Fourier transform of $\hat{T}$. In addition to the super-resolution and phase reconstruction abilities of these algorithms, the forward model can also integrate a digital wavefront correction mechanism (see [7] for further details). This allows an efficient digital focus correction of objects that are out of focus, as is highly desirable for many practical situations.

It is important to observe that the inverse problem to solve is ill-posed. This imposes some experimental conditions on the setup: The thermal noise needs to be sufficiently low as can be achieved by fixing adequate LED power. Furthermore, the number of LEDs (or LR images exploited) should be sufficient. Moreover, and since the descent gradient is performed iteratively, only partial experimental data are exploited (image $I^{i}$) at each step. The overlap factor Γ between two successive regions updated in Fourier Domain at iteration *i* and i+1 must be above 50–60%. This question has been studied by many authors and Γ>60% has been determined from simulations that were further confirmed experimentally [26,27]. These overlapping regions are indicated with the greyed regions in Figure 1. As a result, the Fourier domain needs to be paved with high redundancy, i.e., with many different images $I^{i}$. This limits the throughput of FPM. It is clear that the minimal theoretical quantity of data that should be acquired depends quadratically on the super-resolution factor γ. This γ factor is defined in the Fourier domain by the ratio between the surface to be reconstructed and the surface covered by the numerical aperture of the microscope lens. For example, and for γ=6, only 36 different illumination angles (equivalently images) are theoretically necessary (for γ∼0%). However, in order to ensure a correct convergence of the reconstruction algorithm, Γ should be above 60%. Sample data thus need to be acquired with high redundancy, leading to a penalizing throughput of the microscope. In [7], the experimental configuration used ensures γ = 6 and 225 illuminations were needed. As a result, the minimal time for the acquisition of one FOV in RGB colors is ∼1’30 s considering only 20 ms of exposure time for each stack acquisition (in bright and even dark field conditions).

In the following, we consider an LED circular matrix M (see Figure 1a) since its geometry permits to achieve a low Γ factor. This matrix is split into two distinct groups of LEDs denoted $M_{A}$ and $M_{B}$ with $M=M_{A}\cup M_{B}$. Their layouts are indicated in Fourier domain Figure 1b and Figure 1c, respectively. The LED matrix $M_{A}$ permits us to pave a wide spectral region and hence to attain an important super-resolution factor. $M_{A}$ is composed of 37 LEDs and leads to an overlap factor Γ∼10%. $M_{A}$ geometry has been designed to achieve the lowest overlap factor without any missing information in the Fourier domain. $M_{B}$ is composed of 93 complementary LEDs. When combined with the LEDs of $M_{A}$, they permit us to attain Γ=60% (refer to Figure 1’s caption for further details on the microscope configuration). As already indicated, the use of $M_{A}$ alone cannot permit us to achieve exploitable FPM HR image reconstruction using ePIE or related reconstruction algorithms because of Γ. This property is most likely general. Moreover, we observed (simulations not presented here) that the situation is unchanged even if a global gradient-descent algorithm [28] is used. That means that novel algorithmic reconstruction strategies are needed. We have therefore developed an approach relying on DNNs in order to overcome current limitations.

## 3. Physics-Informed cDIP-DNN Reconstruction Scheme

Our approach relies on a DIP implementation of the reconstruction, seen as an inversion of the forward FPM model described in Section 2.

DIP is a formalism that has been initially introduced in the context of problem inversion in image processing. It relies on an untrained neural network whose weights are optimized for each example to inverse. This approach is radically different from classical learning, as it does not require the use of a training set. It is only interested in solving the inverse problem attached to a single example by including explicitly the forward model equations in a deep neural network framework. The neural network (typically a CNN, an encoder-decoder architecture, or a U-Net [29]) takes as initial input a random noise δ. Its structure provides an implicit regularization during the search for a solution, which allows us to find an adequate solution for ill-posed problems. The objective is to find a set of causes *X* of a phenomenon from the experimental observations of its effects *Y*. We will denote F the forward model linking *X* to *Y*. Typically, *X* is obtained from *Y* by solving an optimization process in the whole space of *X*, which consists of finding the element that best corresponds to the observed effects, once treated by the forward model F. This formulation is often insufficient for the resolution of ill-posed problems and the introduction of additional regularization terms is necessary to constrain the possible *X* values. However, these regularization terms are often ad hoc and may be insufficient for solving ill-posed problems [15]. The interest of DIP networks lies in their capability of introducing another type of regularization based on the weights of the network itself.

DIP approaches have been first employed for solving well-posed problems such as denoising or inpainting problems. They were later extended to solve ill-posed inverse problems where they provide better results than classical methods on both simulation and experimental data [16].

DIP is hence a good candidate for FPM reconstruction. Indeed, we have implemented a DIP approach with a large number of LEDs (Γ>60%) and obtained a good FPM reconstruction in accordance to [30]. Nevertheless, our first experiments revealed that the quality of reconstructed images is rather poor when a limited number of LEDs is employed (i.e., low overlap), similar to EPRY reconstruction. For these reasons, we propose to incorporate an additional learning step in the model prior to the DIP optimization. In such a situation, the weights $\Theta_{s}$ of the network are optimized for a large number of examples. The aim of this first step is to approximate the inverse function itself (from various examples) rather than solving the inverse problem for one specific example. After the convergence of this learning step, a second step is undertaken. There, the inverse function (through the weights $\Theta_{s}$) is solved specifically for the considered data that needs to be inverted. The approximated inverse function obtained from the learning step is used to initialize the optimization step (Figure 2). The same (i.e., unified) cDIP architecture is exploited during these two calculation steps, which ensures a global coherence between the learning step and the optimization one.

### 3.1. Reconstruction via cDIP

The architecture of the DIP model is shown in (Figure 2). The model is composed of two blocks, namely a U-Net and a forward FPM block. The U-Net takes as input the complete stack of experimental Low Resolution (LR) images captured with LEDs of $M_{A}$. Its output is a couple of intensity and phase images or equivalently the researched function $\hat{T}$ of Equation (Equation 5) with $\hat{T}=F\left[I^{\prime }e^{j\Phi^{\prime }}\right]$). The role of the U-Net is to extract information from the images presented at its input. It provides at its output the optimal approximation of the researched solution by minimizing Equation (Equation 6). The U-Net performs different convolutions and pooling operations on several layers and its precise architecture is detailed in Appendix A. Furthermore, the guessed solution $\hat{T}$ that is updated at each epoch is passed to the forward FPM model. This block is in charge of calculating the raw images attached to the output of the U-Net, using Equation (Equation 5). To find the researched solution, one then needs to minimize the MSE between these images and the ones that have been experimentally captured. The resulting loss can be written as
(7)$$L=\sum_{i=1}^{N}{\left|\sqrt{\left|F^{-1}\left[F\left[I^{\prime }e^{j\Phi^{\prime }}\right]\left(k_{x},k_{y}\right)\cdot \hat{C}\left(k_{x}-k_{x}^{i},k_{y}-k_{y}^{i}\right)\right]\right|}-\sqrt{I^{i}\left(x,y\right)}\right|}^{2}$$
where *F* and $F^{-1}$ are, respectively, the Fourier and inverse Fourier transforms, $I^{\prime }$ and $\Phi^{\prime }$ represent the CNN’s intensity and phase output, $\hat{C}$ is the Fourier transform of the point spread function, $k_{x}^{\left(i\right)}$ and $k_{y}^{\left(i\right)}$ stand for the projections of the k-vector along the x- and y-axes, corresponding to the *i*th LED illumination, and $I^{i}\left(x,y\right)$ designates the low-resolution images.

The introduction of the equations describing the microscope image formation in the loss function (Equation (Equation 7)) permits us to consider the model as physics-informed in accordance with [19].

In practice, the weights of the network are initialized randomly and optimized through an iterative process aiming at reducing the loss thanks to gradient descent. The process is stopped when the stability of the loss function is obtained.

As explained before, we have also implemented a scheme that relies on a learning dataset as is described in Section 3.2.

### 3.2. U-Net Learning Scheme for cDIP Initialization (Step 1)

The forward electromagnetic model F attached to Equation (Equation 5) is applied to each HR image (I,Φ) of a learning dataset set (LDS) leading to numerous stacks of LR images formed at camera plan for the different illuminations used (see Figure 3). The size of each stack depends on the number of LEDs considered. More precisely, at this stage, we apply F considering the N = 130 LEDs composing M. We split the resulting LR images into two sub-stacks: those related to the LEDs of $M_{A}$ ($N_{A}$ = 37 images) and to $M_{B}$ ($N_{A}$ = 93 images), respectively.

Indeed, in this learning stage, we take benefit of the availability of these two sub-stacks (namely the reduced stack and the complementary stack) of LR-images to facilitate the convergence of the network and therefore to obtain a good reconstruction quality.

The $N_{A}$ images corresponding to the reduced-led configuration $M_{A}$ are the input of a U-Net, which predicts a couple $\left(I^{\prime },\Phi^{\prime }\right)$ of HR images. Its parameters $\Theta_{s}$ (the weights) are learned, using a DIP optimization of a loss function, which is the mean squared error between the initial N images stack and the N images stack resulting from the forward model applied on $\left(I^{\prime },\Phi^{\prime }\right)$. At each iteration *i*, a novel couple of images of the learning dataset is considered; in this way, the parameters of the U-Net, the $\Theta_{s}$, are optimized for a large number of examples, contrary to what occurs in the classic DIP implementation where only one example is considered.

Note that, even if the input of the U-Net is limited to the reduced stack of images, the loss function exploits the complete stack of N images. In this way, the final reconstructed image benefits from rich information obtained from the original HR images. Furthermore, the loss function does not consider an MSE between (I,Φ) and $\left(I^{\prime },\Phi^{\prime }\right)$ as is generally carried out in the literature. Indeed, in the current approach, the loss function results from the comparison of the LR images resulting from the forward model with the complete number of LEDs N, applied to both couples of HR images. This guarantees that the physical equations are explicitly solved during the learning step. We denote cDIP-L as the model of step 1 (for the cDIP-Learned model).

### 3.3. cDIP Reconstruction under Low Overlap Conditions (Step 2)

Once the parameters of the network are learned (after a suitable number of iterations), the model is ready to reconstruct an unknown stack of FPM LR-images acquired under reduced overlap Γ conditions. The reduced stack of $N_{A}$ images (related to illuminations $M_{A}$) is presented at the input of the U-Net. As explained before, we have a last step (step 2) in order to refine the solution. This is obtained by solving the forward model that is attached to one single set of LR images only, as detailed in the Section 3.1. It is to be noted that such optimization is performed without any change in the architecture and the same loss function that was introduced in step 1 is exploited. The only difference lies in the fact that the weights of the network $\Theta_{learned}$, obtained after the training phase, are used for the initialization of the U-Net. Also, these weights are now optimized using the reduced stack only since it is the only information available at this stage. The final reconstructed HR image is finally obtained at the output of the network after the convergence of the optimization step.

We note that the interest of our approach is that the models used in the learning and optimization phase are identical. The two steps are hence unified from DNN model point of view. In the following, this model is denoted cDIP-LO (for cDIP learned and optimized model) when the model is initialized with learned weights and cDIP-O when the initialization of DNN weights is random only.

## 4. Simulation Results and Discussion

The microscope configuration that is employed to evaluate the cDIP-LO reconstruction model consists of a microscope fitted with an objective lens of magnification 4× and NA = 0.08. The model is tested with simulations only to evaluate its ultimate performance.

The LED matrix M composed with 130 LEDs at λ=525 nm introduced in Figure 1 is placed at a distance d = 55 mm below the sample. The different regions of the sample spectrum that are consequently probed correspond to the many regions already indicated in Figure 1b,c. Such configuration allows us to reach a theoretical super-resolution factor γ=5. The pixel pitch of the camera that is simulated is 3.45
μm. The initialization weights of the cDIP-LO model are obtained using the Learning DataSet LDS (cf. Figure 3) after training the cDIP-L model. LDS is constructed from ImageNet Large Scale Visual Recognition 2019 Challenge (ILSVRC’19) and contains 10,000 complex masks $T_{j}$ (refer to Equation (Equation 1)). Each of them represents a numerical sample used to simulate FPM LR-images. More precisely, $T_{j}$ is obtained from images taken in the ImageNet database. For each *j* (*j* varying from 1 to 10,000), two images of 384 × 384 pixels (respectively $I_{j}$ and $\Phi_{j}$) are randomly extracted from the ImageNet catalog. $T_{j}$ is then formed with $\sqrt{I_{j}}e^{j\Phi_{j}}$. For each $T_{j}$, a stack of 130 FPM LR-images simulating the photos that would be acquired by the camera on sample $T_{j}$ is calculated using a forward model (see Equation (Equation 5)) for the different LEDs of the illumination matrix $M=M_{A}\cup M_{B}$. The overlap factor Γ is ∼60%. It is important here to highlight that the LDS that is thereby constructed permits us to simulate a vast variety of samples that would hardly be accessible to experimental FPM measurements. In particular, ImageNet contains a considerable amount of photos with wide variations in their spatial characteristics (contrast, spatial frequencies, etc.). Also, *I* and Φ functions are uncorrelated by construction. This is usually not the case in real experiments since biological samples absorption and optical thickness are generally closely related to one another. Such LDS could therefore be beneficial for good model generalization. Because of the camera pitch used in the simulations (of 3.45
μm) and the objective lens characteristics (4× NA = 0.08), the produced LR images are sampled by the camera with an important oversampling factor with respect to Shannon criteria. In our case, each LR image is 128 × 128 pixels. In complement to LDS, a test dataset TDS is also constituted. It consists of 1000 complex masks that are obtained following the same construction that is used for the LDS. The images taken from ImageNet are however different than the one used in LDS. For each mask, two stacks of LR images are also calculated using FPM forward model. The first stack is related to LEDs of $M_{A}$, and the second to LEDs of M.

The TDS is exploited to compare the different reconstruction models (ePIE, cDIP-L, and cDIP-LO). A typical result is presented in Figure 4, where images (a) and (g) correspond to *I* and Φ HR-images that are reconstructed with the ePIE algorithm using the stack of LR-images relative to LEDs of M. These two images constitute the targeted (or reference) intensity and phase HR images. Image (b) corresponds to the raw image related to the matrix central LED. There, the native resolution of the objective lens can be appreciated. The other images in Figure 4 are the images that are reconstructed using the LR-images stack relative to LEDs of $M_{A}$ only. In particular, images (c) and (h) correspond to ePIE reconstruction, images (d) and (i) to cDIP-L reconstruction, and (f) and (j) to cDIP-LO (intensity and phase). Images (c) and (h) illustrate the typical artifacts that can be obtained with ePIE reconstruction under low overlap conditions. Important crosstalk between the intensity and phase is observed. Also, the phase contrast is highly degraded. It is clear that the quality of these images is too low to consider their exploitation for real applications (with low Γ). In contrast, *I* and Φ images obtained with cDIP-L are of improved perceptual quality both for intensity and phase. However, artifacts are still noticeable. Some of them are pointed out with arrows in Figure 4. For example, arrow A corresponds to degradations in the high frequencies that are mostly lost. Arrow B corresponds to an incomplete separation between phase and intensity image (crosstalk). Although cDIP-L permits us to recover phase information with benefits as compared to ePIE, the final resolution of reconstructed images is questionable: the intensity image resolution is almost comparable to the raw intensity image relative to central LED illumination. This is consistent with state-of-the-art published results [11], also relying on trained systems. This seems to be inherent to models that are only statistically estimated. Interestingly, the situation is much different with the cDIP-LO model. The differences between reconstructed and reference images are too low to be visually appreciable.

This improvement brought by cDIP-LO neural reconstruction is further confirmed by evaluating its performances quantitatively with the error function L that is calculated over the complete TDS dataset. The traditional SSIM and PSNR metrics are evaluated for reconstructed images with respect to reference images for intensity and phase. The results are indicated in Table 1 where the mean and standard deviation of L, SSIM, and PSNR are calculated on the whole TDS. The ePIE column designates ePIE reconstruction using LEDs of $M_{A}$. Unambiguously, the cDIP-LO algorithm leads to better image reconstruction. In particular, the loss evaluated for ePIE and cDIP-L is almost comparable, whereas it is 10 times lower for cDIP-LO. The similarity metric SSIM is close to 1 for cDIP-LO (as desired) and much higher than cDIP-L and ePIE SSIM. The PSNR also reveals an important increase for cDIP-LO compared to cDIP-L and ePIE (above 10 dB).

It is hence clear that the model cDIP-LO improves substantially the reconstruction performances. We attribute such improvement to the fact that the initialization of cDIP-LO weights benefits information that is relative to all the LEDs of M during the learning step and the fact that cDIP-L reconstruction calculations are further solved using forward model equations. One can also observe from Table 1 that standard deviations of metrics relative to cDIP-LO are low, indicating the ability of this DNN approach to generalize. We recall that TDS has been constituted with various artificial complex masks with no correlation between intensity and phase precisely.

The metrics L, SSIM, and PSNR do not provide information regarding the final delivered image resolution. For this reason, we have undertaken complementary simulations of USAF 1951 intensity target resolution chart reconstruction using LEDs $M_{A}$. Typical results are presented in Figure 5. The first line of images corresponds to the situation where the USAF1951 object is placed at the microscope focal plane. For the second line, the object is out of focus and placed 80 μm above. Figure 5a,b correspond, respectively, to the reference image and the raw image relative to the central LED. The last element that is resolved is indicated with an arrow. The reconstructions obtained with ePIE, cDIP-L, and cDIP-O (i.e., without initialization of the cDIP model) are shown in images Figure 5c–e, respectively. As can be observed, the last resolved element does not change much. That means that no significant improvement in resolution (as compared to native objective lens resolution) is observable with reconstruction, although the global perceptual quality of the reconstructed images is improved. In contrast, cDIP-LO reconstruction (see Figure 5f) reveals a significant change in the position of the last element that is resolved. It permits us to evaluate that the super-resolution factor that is attained is γ=5, as is theoretically expected. Note also that no appreciable difference with respect to the reference image is observable.

Although the perceptual quality of the USAF1951 object is minimally a little enhanced as soon as reconstruction is performed (compare Figure 5c–f) to image Figure 5b, the situation is much different when the object is out of focus (second line of images of Figure 5). The images of Figure 5i–l correspond to images reconstructed with digital focus correction (as explained in Section 2). One can note that the quality of images obtained with ePIE, cDIP-L, and cDIP-LO is degraded. In contrast, cDIP-LO demonstrates its ability to reconstruct correctly the resolution target chart without any degradation. This reconstruction algorithm is hence fully functional and is able to handle correctly the integration of focus correction in the forward model, as is highly desirable for practical situations.

## 5. Conclusions

In this paper, we have introduced a DNN-based algorithm, namely cDIP-LO DNN, to reconstruct images acquired with FPM under a reduced number of illuminations of the sample (i.e., Γ<60%). Reconstruction properties of such an approach have been compared to the state-of-the-art algorithms ePIE and to the statistically learned DNN model. The simulated results for an FPM configuration with a very low Γ factor (Γ∼10%) demonstrate the conceptual benefits of cDIP-LO in terms of image quality and resolution. The specificity of cDIP-LO is two-fold: (1) it is a physics-informed DNN, in the sense that the forward image microscope formation model is explicitly introduced in the model to calculate the loss function; the HR reconstructed images are therefore obtained by this model, but their values have never been used directly to calculate the loss, (2) the cDIP-LO model’s weights are initialized after a statistical learning step; in this step, the forward model has been purposely introduced in the replacement of any exploitation of a reference solution. This allows the employment of a single and coherent model (or unified DNN architecture) in the learning step and in the optimization step. We note that the learning step is mandatory; the reconstruction with cDIP-LO without initialization of the model leads to unexploitable images.

In future work, the test of cDIP-LO on experimental data will be undertaken. From a practical point of view, we believe that simulated data should be sufficient in the learning step without any need for experimental data. This point will be specifically studied with respect to experimental noise tolerance or possible LEDs position imprecision. The idea of coupling a statistical learning step and a physics-informed DNN optimization is general. It is readily applicable to the classic commercial LED rectangular board. It can also open interesting perspectives for FPM reconstruction under multiplex-coded illumination or for diffraction tomography.

## Figures and Tables

**Figure 1 sensors-23-06829-f001:**
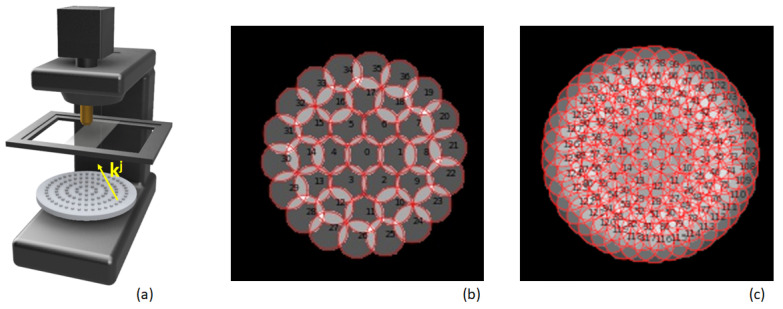
FPM reconstruction in Fourier domain. Schematic presentations of the microscope fitted with a LED matrix $M=M_{A}\cup M_{B}$ (**a**), spectral region covered in Fourier Domain with $M_{A}$ (**b**) and with $M_{B}$ (**c**).

**Figure 2 sensors-23-06829-f002:**
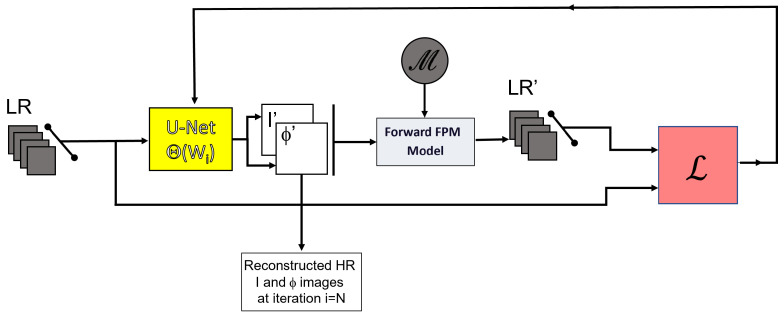
cDIP model: LR designates the stack of FPM low-resolution images measured at camera plan using LEDs M. Θ represents the parameters of the U-Net that are optimized after each iteration of backpropagation calculations. $I^{\prime }$ and $\Phi^{\prime }$ represent the HR images reconstructed with U-Net at each iteration. The reconstructed images I,Φ are obtained when the model has converged. LR’ is the stack of low-resolution images calculated by the forward model. The loss function L is defined in Equation (Equation 6).

**Figure 3 sensors-23-06829-f003:**
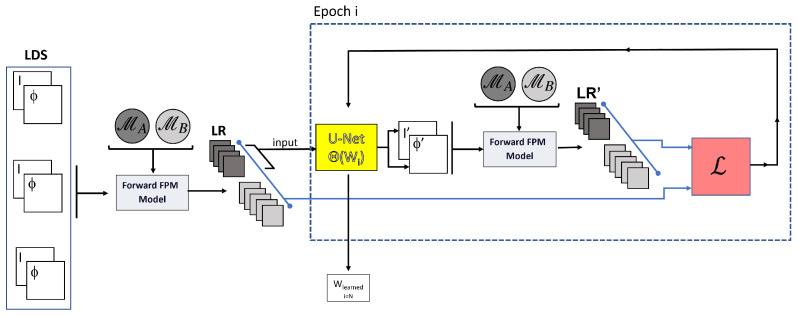
cDIP-L model (U-Net learning model). LDS: learning dataset composed with many intensity *I* and phase Φ images couples. The forward FPM model is used to calculate low-resolution images formed at the camera plan. $M_{A}$ and $M_{B}$ represent the splitting of the LED matrix into two groups. LR: stack of low-resolution images formed at camera plan for the different illuminations used. Θ represents the parameters of the U-Net. LR’: low-resolution images calculated from the U-Net predicted images $I^{\prime }$ and $\Phi^{\prime }$ (in dark gray for $M_{A}$, light gray for $M_{B}$). L: loss function.

**Figure 4 sensors-23-06829-f004:**
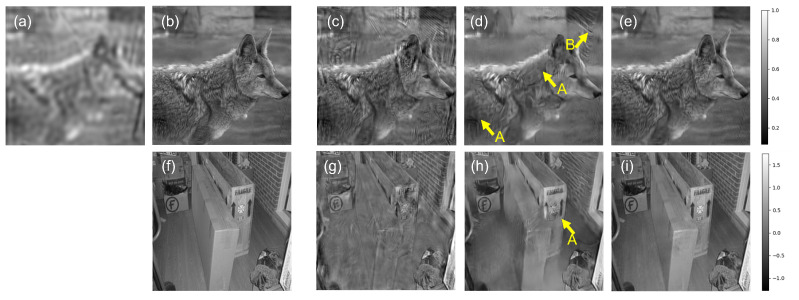
Comparison of reconstruction results obtained with illuminations with an overlap ∼10%. Images (**a**–**e**) correspond to intensity images, whereas images (**f**–**i**) to phase images. (**a**) Raw image acquired with central LED, (**b**) reference image, (**c**) ePIE reconstruction (using LEDs $M_{A}$), (**d**) cDIP-L reconstruction after training (using LEDs $M_{A}$), (**e**) cDIP-LO reconstruction after training and optimization (using LEDs $M_{A}$). (**f**) corresponds to reference phase, and (**g**–**i**) to reconstructed phase with e-PIE, cDIP-L and cDIP-LO, respectively. The arrows indicate different types of artifacts (such as crosstalk or resolution degradation). References images (**b**,**f**) are obtained from ePIE reconstruction using all stack of images (LEDs M).

**Figure 5 sensors-23-06829-f005:**
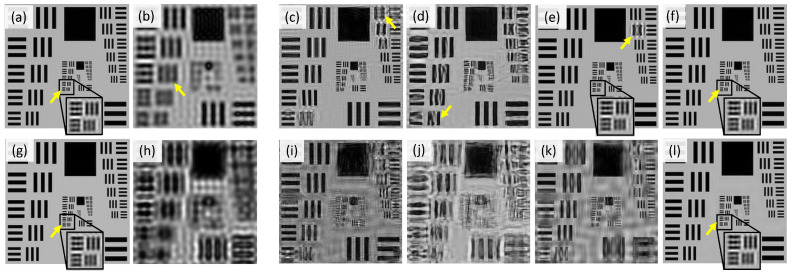
Comparison of intensity USAF 1951 target resolution chart reconstructions obtained with illumination $M_{A}$ (overlap ∼10%). For images (**a**–**f**), the target resolution chart is placed at the microscope focus plan. (**a**) Reference, (**b**) raw image acquired with central LED illumination, (**c**) ePIE, (**d**) cDIP-L, (**e**) cDIP-O, and (**f**) cDIP-LO reconstructions. For images (**g**,**h**), the resolution chart is placed 8 μm above the microscope focus plan. (**g**) Reference (identical to image (**a**)), (**h**) central LED, (**i**) ePIE, (**j**) cDIP-L, (**k**) cDIP-O, and (**l**) cDIP-LO reconstructions. The reference is calculated using ePIE reconstruction with M LEDs illumination.

**Table 1 sensors-23-06829-t001:** L, SSIM and PSNR metrics evaluated over the complete TDS dataset. The metrics’ mean values and standard deviations are indicated. Reconstructions with the different algorithms (ePIE, cDIP-L, and cDIP-LO) are obtained using the stack of LR images obtained with LEDs of $M_{A}$.

	ePIE	cDIP-L	cDIP-LO
$L\times (10^{-4})$	$\bar{\mu }=4.43$	$\bar{\mu }=1.47$	$\bar{\mu }=0.244$
	σ=2.85	σ=0.724	σ=0.0941
$SSIM_{Intensity}$	$\bar{\mu }=0.569$	$\bar{\mu }=0.671$	$\bar{\mu }=0.919$
	σ=0.143	σ=0.105	σ=0.029
$SSIM_{Phase}$	$\bar{\mu }=0.19$	$\bar{\mu }=0.491$	$\bar{\mu }=0.915$
	σ=0.109	σ=0.167	σ=0.06
$PSNR_{Intensity}$ (dB)	$\bar{\mu }=17.28$	$\bar{\mu }=25.03$	$\bar{\mu }=34.8$
	σ=4.38	σ=2.68	σ=2.71
$PSNR_{Phase}$ (dB)	$\bar{\mu }=18.32$	$\bar{\mu }=25.05$	$\bar{\mu }=39.14$
	σ=3.53	σ=1.86	σ=2.81

## Data Availability

The corresponding author can be contacted for access to the data presented in this study upon request. The data are not publicly available due to restrictions on privacy.

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
