# Peer review of "A Physics-Inspired Deep Learning Framework for an Efficient Fourier Ptychographic Microscopy Reconstruction under Low Overlap Conditions"

_sensors, 2023, doi:10.3390/s23156829_

Round 1
Reviewer 1 Report
The manuscript has been carefully revised based on the reviewer's comments, and all the comments have been addressed properly. In my opinion, the paper can be accepted in the form.
Minor editing of English language required
Author Response
Dear Reviewer,
We would like to thank you for your feedback on the manuscript.
Reviewer 2 Report
In this work, the authors provide cDIP-LO DNN, a DNN-based approach for reconstructing images taken with Fourier Ptychographic Microscopy (FPM) under a limited number of sample illuminations.
The main issue with this manuscript is that the authors do not explain the main features which is the CNN used in this study. Moreover, it is not clear how the method become physics-informed. These are the important aspects of the manuscript and must be explained elaborately. I would be happy to review this manuscript once the authors incorporate the suggested changes.
See above
Author Response
Response to Reviewer 2 Comments
Dear Reviewer,
We would like to thank you for your useful comments to improve the manuscript.
Point 1: The main issue with this manuscript is that the authors do not explain the main features which is the CNN used in this study. Moreover, it is not clear how the method become physics- informed. These are the important aspects of the manuscript and must be explained elaborately. I would be happy to review this manuscript once the authors incorporate the suggested changes.
Response 1:
We have clarified the type of neural network we have used, i.e. a U-Net. This allows us to be more precise rather than talking about a CNN as in the first version. Details of the U-Net architecture used for image reconstruction are now introduced in Appendix A. It seemed more appropriate to place these details and main features in an appendix, firstly so as not to interrupt the reading of the article, and secondly because this architecture is classic and well documented.
To clarify the Physics-informed approach from the neural approach, a new section has been introduced (cf. section 3.1, line 218 of the red-lined version of the article). It focuses on the cost function. The function that is used results from the equations attached to the forward model of the FPM microscope (those introduced in section 2). The equation that is minimized in the DIP model is therefore identical to the one minimized in conventional FPM reconstruction algorithms and dictated by the physical model used to describe the image formation of the sample being reconstructed.
The fact that the physical laws are explicitly introduced into the cost function (information prior) of the DNN, and the fact that it is minimized for a specific set of LR images (one example), renders the approach Physics Informed, in line 72 with the reference [19].
It should be noted, however, that rather than using a random initialization of the network as is classically done in DIP approaches, here the initialization weights of the U-Net are learned from a database as detailed in section 3.2.
To facilitate the reading of the article and based on the questions raised by the reviewer, we have retained the following organization:
Section 3.1: DIP optimization architecture at the heart of the neural approach implemented (DIP).
Section 3.2: Learning initialization weights using examples
Section 3.3: DIP reconstruction by optimization with previously learned initialization.
In addition, the article has been carefully proofread to improve as much as possible the English language where necessary.
Reviewer 3 Report
The paper “A physics-inspired deep learning framework for an efficient FPM reconstruction under low overlap conditions” introduces a new deep learning method for super-resolution of Fourier Ptychographic Microscopy under low overlap conditions. The proposed method demonstrates effectiveness in their experimental evaluation, showcasing its potential for enhancing FPM imaging.
However, I have the following major concerns,
(1) Where is the physics involved. Based on their design I think in the first step, cDIP-L model part, the model learns the physics which is encoded in the Forward FPM model based on Deep Image Prior (DIP) networks. This is still the learned model and does not directly encode physics into their model.The authors could provide further clarification on the specific aspects of physics-inspired techniques and reference the Nature Review on Physics-informed machine learning for additional context,Physics-informed machine learning https://www.nature.com/articles/s42254-021-00314-5.
(2) The key benefits of Deep Image Prior (DIP) networks are that it typically does not require paired training examples and prevent overfitting. However, in the proposed method, the authors utilize higher-resolution and lower-resolution pairs for training. It would be valuable for the authors to elaborate on why this approach is necessary and how it provides advantages compared to traditional DIP networks, clarifying the specific benefits and improvements achieved through this training setup.
(3) More quantitative metrics are needed in addition to SSIM to comprehensively characterize the super-resolution results. Figure 4 demonstrates the importance of incorporating various metrics, and it would be beneficial for the authors to provide a wider range of evaluation metrics to assist readers in accurately assessing the performance of their methods.
Some small errors,
-
In Line 71, "The DIP network is fully physics-informed across the 2 steps in the sens that exact forward model of" seems to contain a typo. It should be "sense" instead of "sens".
-
In Line 92, "source is replaced by a led matrix array (see fig. 1a)." The word "led" should be capitalized as "LED" since it stands for light-emitting diode.
-
In Line 272, "that is implemented are indicated in annex 1." I couldn't find "annex 1" in the document.
-
In Line 221, LDS is referred to as "a learning dataset base (LDS)" but in Line 274, the author uses "a Learning DataSet (LDS, cf. fig. 2)". Please unify the capitalization of notations to ensure consistency throughout the paper.
Please proofread the text to address any typos and simplify the sentences to improve the readability of the English text.
Author Response
Response to Reviewer 3 Comments
We would like to thank reviewer #3 for his useful comments to improve the manuscript.
Point 1: Where is the physics involved. Based on their design I think in the first step, cDIP-L model part, the model learns the physics which is encoded in the Forward FPM model based on Deep Image Prior (DIP) networks. This is still the learned model and does not directly encode physics into their model. The authors could provide further clarification on the specific aspects of physics-inspired techniques and reference the Nature Review on Physics-informed machine learning for additional context, Physics- informed machine learning https://www.nature.com/articles /s42254-021-00314-5.
Response 1:
To clarify the Physics-informed approach from the neural approach, a new section has been introduced (cf. section 3.1, line 218 of the red-lined version of the article). It focuses on the cost function. The function that is used results from the equations attached to the forward model of the FPM microscope (those introduced in section 2). The equation that is minimized in the DIP model is therefore identical to the one minimized in conventional FPM reconstruction algorithms and dictated by the physical model used to describe the image formation of the sample being reconstructed.
The fact that the physical laws are explicitly introduced into the cost function (information prior) of the DNN, and the fact that it is minimized for a specific set of LR images (one example), renders the approach Physics Informed, in line 72 with the reference [19] kindly suggested by reviewer#2.
Point 2: The key benefits of Deep Image Prior (DIP) networks are that it typically does not require paired training examples and prevent overfitting. However, in the proposed method, the authors utilize higher-resolution and lower-resolution pairs for training. It would be valuable for the authors to elaborate on why this approach is necessary and how it provides advantages compared to traditional DIP networks, clarifying the specific benefits and improvements achieved through this training setup.
Response 2:
The images used for training are for model initialization only. They are made once and for all. It's true that this learned model could also be used as a model for image reconstruction (inference). This way of reconstructing data has already been discussed in the literature and does not give complete satisfaction at high resolutions [11,12]. These previous results have precisely motivated this work, as explained in the introduction [cf. line 51-54].
To address concerns regarding the necessity of the initialization part of the optimized cDIP model, and as indicated in lines 206-207 (red-lined version), we noticed that the reconstruction without initialization and when the number of LEDs is reduced did not allow the phase to be reconstructed correctly. This can be seen in Figure 5.
The learning step that is here introduced, allows us to use the complete information of all the LEDs through the average network weights that are consequently calculated from the database under high overlap conditions. As a result, the optimization step starts from an initial solution that is close to the desired one leading to an adequate convergence even when the overlap condition is low.
Point 3: More quantitative metrics are needed in addition to SSIM to comprehensively characterize the super-resolution results. Figure 4 demonstrates the importance of incorporating various metrics, and it would be beneficial for the authors to provide a wider range of evaluation metrics to assist readers in accurately assessing the performance of their methods.
Response 3:
The PSNR metric, also very common, has been incorporated in table 1. The different metrics are consistent and show a similar trend. We believe that SSIM and PSNR are the 2 most widely used metrics in the literature.
Point 4 :Some small errors,
- In Line 71, "The DIP network is fully physics-informed across the 2 steps in the sens that exact forward model of" seems to contain a typo. It should be "sense" instead of "sens".
- In Line 92, "source is replaced by a led matrix array (see fig. 1a)." The word "led" should be capitalized as "LED" since it stands for light-emitting diode.
- In Line 272, "that is implemented are indicated in annex 1." I couldn't find "annex 1" in the document.
- In Line 221, LDS is referred to as "a learning dataset base (LDS)" but in Line 274, the author uses "a Learning DataSet (LDS, cf. fig. 2)". Please unify the capitalization of notations to ensure consistency throughout the paper.
Response 4:
All errors pointed out by reviewer no.3 have been corrected in the text. In addition, the article has been carefully proofread to improve as much as possible the English language where necessary
Round 2
Reviewer 2 Report
In the revised version, it is still not clear how the equations in Section 2 are added to the cost function to make the DNN physics-informed. The author should clarify this by explicitly writing the cost function.
Author Response
Response to Reviewer 2 Comments
We would like to thank reviewer #2 for his useful comments to improve the manuscript.
Point : In the revised version, it is still not clear how the equations in Section 2 are added to the cost function to make the DNN physics-informed. The author should clarify this by explicitly writing the cost function.
Response :
In order to clarify in which way our DNN is physics-informed, we have detailed the cost function in Equation 7. Indeed, as advised by reviewer #2, we have introduced the equations of section 2 related to the physical image formation model into the loss function.
We hope that this clarifies this point.
Reviewer 3 Report
The revised version looks good to me.
Author Response
Dear Reviewer,
We would like to thank you for your positive feedback on the manuscript.
Round 3
Reviewer 2 Report
Accept in present form